# MicroRNAs and ‘Sponging’ Competitive Endogenous RNAs Dysregulated in Colorectal Cancer: Potential as Noninvasive Biomarkers and Therapeutic Targets

**DOI:** 10.3390/ijms23042166

**Published:** 2022-02-16

**Authors:** Brian G. Jorgensen, Seungil Ro

**Affiliations:** Department of Physiology & Cell Biology, University of Nevada, Reno School of Medicine, Reno, NV 89557, USA; brianjorgensen@med.unr.edu

**Keywords:** colorectal cancer, miRNA, ceRNA, lncRNA, circRNA, sponging, noninvasive biomarkers

## Abstract

The gastrointestinal (GI) tract in mammals is comprised of dozens of cell types with varied functions, structures, and histological locations that respond in a myriad of ways to epigenetic and genetic factors, environmental cues, diet, and microbiota. The homeostatic functioning of these cells contained within this complex organ system has been shown to be highly regulated by the effect of microRNAs (miRNA). Multiple efforts have uncovered that these miRNAs are often tightly influential in either the suppression or overexpression of inflammatory, apoptotic, and differentiation-related genes and proteins in a variety of cell types in colorectal cancer (CRC). The early detection of CRC and other GI cancers can be difficult, attributable to the invasive nature of prophylactic colonoscopies. Additionally, the levels of miRNAs associated with CRC in biofluids can be contradictory and, therefore, must be considered in the context of other inhibiting competitive endogenous RNAs (ceRNA) such as lncRNAs and circRNAs. There is now a high demand for disease treatments and noninvasive screenings such as testing for bloodborne or fecal miRNAs and their inhibitors/targets. The breadth of this review encompasses current literature on well-established CRC-related miRNAs and the possibilities for their use as biomarkers in the diagnoses of this potentially fatal GI cancer.

## 1. Introduction

As a generalized class of biomolecules, RNA species occupy a unique niche that can alternate between being transient and/or enduring, while being consequential or seemingly neutral to cellular homeostasis, growth, and survival across a myriad of taxonomic levels of life. While RNA has been known as a unique biomolecule since the 19th century, it came to the forefront of biological study in the mid-20th century upon the discovery that various RNA molecules were, collectively, the transcript, delivery, and production scaffolding for translating the genetic information contained within DNA into functional proteins via ribosomes. However, in the decades since those foundational studies, it has been found that a vast majority of the biological activity of RNA species is not as transcript templates (messenger RNA (mRNA)) for protein synthesis, but rather regulators of gene expression. These non-mRNA RNA transcripts do not primarily function as templates for protein translation and, thus, are called noncoding RNAs (ncRNA). One important class of endogenous regulatory ncRNAs are microRNAs (miRNA), which are 21–25 nucleotides in length when mature following enzymatic processing. Following their initial discoveries as endogenously produced transcripts, it was found that the dysregulation of many specific ncRNAs are tied to malignancies in humans [1,2,3,4]. Subsequent studies into each individual/category of ncRNA built upon these discoveries of dysregulated ncRNA expression to reveal the modes of action on cellular pathways known to be causative in carcinogenesis. Certain miRNAs were found to be key regulators of well-established carcinogenic pathways such as miR-143 and miR-145, acting as tumor suppressors on the p53/c-Myc pathway [5,6], the circular RNA (circRNA) Cdr1as sequestering miR-7, reducing its ability to regulate the proto-oncogenic PI3K/AKT pathway [7,8], and the long noncoding RNA (lncRNA) GAS5-controlling mTOR-mediated proliferation via the competitive binding of glucocorticoid receptors and miR-21 [9,10]. While these miRNAs can affect many distinct cellular pathways, there is oftentimes an overlap and/or interplay between unique ncRNAs classes that results in the pathological alterations of many signaling pathways at the cellular level. This can manifest, for example, when lncRNAs or circRNAs competitively sequester miRNAs from their targeted mRNAs (collectively known as competitive endogenous RNAs, ceRNA) through complementary base pairing, known as “sponging”, as seen between the long noncoding RNA (lncRNA) HOTAIRM1 or the circular RNA (circRNA) circITCH and miR-17 [11,12]. Colorectal cancer (CRC) most commonly manifests as neoplasias consisting mainly of mutated intestinal epithelial cells that are unable to maintain a proper differentiation status and/or connections to neighboring cells, leading to unchecked cellular division and the dysregulation of oncogenic and/or tumor-suppressive genes [13,14]. Most CRC cases do not yet have a prescribed causative provenance and are, thus, considered sporadic (70%) [15]. For decades, invasive colonoscopies have been the most common form of early detection for CRC and they luckily reduce the death rate by about 50% [16,17]. Unfortunately, this reduction in mortality appears to be largely associated with carcinomas found on the left side of the colon as they are easier to observe and remove [18], leaving around 10% of all CRCs undetectable via colonoscopy [19]. This inability to robustly detect CRC regardless of the location provides the impetus to find biomarkers, such as miRNAs, in excreted bodily fluids (stool/serum/plasma) that could potentially diagnose CRC in various colonic regions. These strategies of measuring levels of known oncogenes in bodily fluids (mainly serum, plasma, or stool) have already begun to be investigated and employed for miRNA [20,21,22], lncRNA [23,24], and circRNA [25,26]. Unfortunately, due to miRNAs primarily exerting direct influence on other RNA molecules or proteins, which may or may not coexist together with the miRNA of interest, the basic presence or absence of any given miRNA is not necessarily enough to diagnose CRC. This conundrum adds a layer of difficulty in attempting to understand the relationship of these miRNAs to GI pathologies resulting in many unclear, or even contradictory, connections between the disease state and miRNA across samples and studies. Thus, to avoid a cavalcade listing of any miRNAs statistically associated with CRC, this review focuses on known miRNA signatures found within the biofluids of CRC patients, that also have accompanying evidence-based mechanistic hypotheses which represent ideal candidates for the use of biomarkers as diagnostics and/or therapeutics.

## 2. miRNAs and Colorectal Cancer

miRNAs are short RNA molecules, only 21–25 nucleotides in length upon the completion of the processing of the stem-loop pre-miRNA via the endoribonucleases Drosha and Dicer into single-stranded mature miRNAs, which are bound by Argonaute proteins for delivery to sequence-specific sites [27]. The most well-established method of the action of mature miRNAs is that of the binding targeted mRNAs through a nucleotide base pair complementarity between the miRNAs seed sequence (nucleotides 2–8 from the 5′ end) and the targeted mRNA sequence, most commonly at the 3′ UTR or coding sequences leading to varied changes in the translation efficiency of the bound mRNA into a protein product [28,29,30,31]. A reduction in translation in the associated protein is most frequently observed, but increases in the associated protein expression do occur [32,33,34]. Additionally, strong evidence suggests the noncanonical binding of miRNAs occurs with functional effects as seen with miR-21 binding/activating Toll-like receptor proteins (TLR8) [35], as well as binding the lncRNA GAS5 [36], with both leading to pro-inflammatory signaling cascades, which are also found in CRC. Cataloging and correlating miRNAs associated with malignant tissue in order to uncover biomarkers has been an intense area of research for decades after it was found that miRNAs are better candidates for biomarker investigation than mRNAs [37,38,39]. Current literature suggests there are well over 230 miRNAs, and likely many more, associated with CRC, with some frequently appearing in various tissues/fluids/cells and others only rarely appearing [20,40,41,42,43,44,45,46,47,48,49,50,51,52,53]. Due to the seemingly boundless nature of connecting miRNAs to CRC tissue/biofluids/cells, the focus of this review is on some of the most well-established miRNAs, their pathways that lead to CRC, and observed levels in biofluids (Table 1, Figure 1) instead of an exhaustive inventory of all potential candidates. Many miRNAs are possible contenders for biomarking CRC in bodily fluids, as miRNAs affect numerous known pathways to malignancy as found below.

### 2.1. miR-21

The MIR21 gene is located on chromosome 17, and was one of the first miRNAs found whose expression was positively associated with cancers, specifically hepatocellular and breast, via microarray [119,120,121], and is naturally highly expressed in immune cells (monocytes, macrophages, and dendritic cells) [122,123]. In fact, miR-21 has been found to be related to at least 29 disease conditions, leading to controversy on using miR-21 as a viable biomarker for specific diseases [124]. The global knockout of miR-21 does not appear to cause any phenotypic pathologies outside of elevated levels of some of its target genes in specific cell types [125]. Genomically, MIR21 is found within the VMP1 locus whose protein product is vital in maintaining cell-to-cell connections and the loss of its expression can lead to aggressive colorectal cancer [126,127]. Recently, it was found that miR-21 indirectly represses the expression of VMP1 through the inhibition of miR-21’s known tumor suppressing target, PTEN, creating a negative feedback loop on VMP1 with increasing miR-21 levels [128]. The canonical pathway for miR-21-induced oncogenesis is through to be the direct repression of various well-known tumor suppressor genes, including PTEN [120,129] and PDCD4 [130,131], which activate cyclin-dependent kinases, c-MYC, and PI3K/AKT/mTOR pathways, which results in an increased invasion and metastasis [10,125,132,133,134]. MiR-21 also induces the increased expression of anti-apoptotic proteins such as BCL2 [121,135], and regulates more than twenty-five other known targets [136]. An analysis of resected colonic tissue confirmed the inverse relationship of miR-21 and PDCD4 in colorectal tumors, and their comparative expression levels can predict metastasis [137,138]. In addition to canonical pathways in CRC, miR-21 is known to bind and activate TLR8 protein [35], as well as binding and inhibiting the anti-inflammatory lncRNA GAS5 [36]. Furthermore, miR-21 levels have consistently been shown to be elevated in both the serum [57,58,60,63,84] and fecal [55,56] samples of CRC patients. MiR-21 is implicated in several unique disease states and, thus, the use of its presence alone as an indicator for CRC is likely only a small piece of future miRNA biomarkers for CRC when combined with the expression levels of other ncRNAs and mRNA targets found in CRC patients.

### 2.2. miR-17/92 Cluster

The miR-17/92a cluster is found within the third intron of the C13ORF25 locus on chromosome 13, contains miR-17, miR-18a, miR-19a, miR-20a, miR-19b, and miR-92a, and was initially found as a polycistronic and oncogenic lncRNA (MIR17HG) in lung cancer cells [139,140]. Every member of the miR-17/92 cluster has been found to be associated with either CRC tissue or plasma/serum [68,141,142,143]. The seed sequences of many of these mature miRNAs are redundant and found in other mature miRNAs. Each miRNA in the miR-17/92 has at least one redundant miRNA from within the miR-17/92a locus or within the miR-106a/363 or miR-106b/25 locus [140]. The deletion of the entire miR-17/92 cluster results in embryos of reduced size, which are fatal immediately after birth and the dual deletion of miR-17/92a and miR-106b/25 causes embryonic lethality, while the singular deletion of either the miR-106b/25 or miR-106a/363 cluster (or their combined ablation) has not produced a similar phenotype [144]. In general, the functional duality of miRNAs being both essential to development and oncogenesis is well-established, as they are both growth-promoting states [145]. In humans, the germline hemizygous deletion of miR-17/92 results in patients with type 2 Feingold syndrome [146]. These results emphasize that the essential nature of the miR-17/92 cluster cannot be rescued with seed sequence redundancy found in other miRNAs at other loci, further implying the importance of noncanonical cellular influence. Similar to many miRNAs, the miRNAs in the miR-17/92 cluster produce their oncogenic effect on several known pathways. In CRC tissue, miR-17 downregulates RBL2, leading to carcinogenic Wnt/ß-catenin induction [147], but has also shown the potential inhibition of colorectal cancer invasiveness when used in isolation [148]. Levels of miR-18a were found to be elevated in the serum of CRC patients and, thus, are a potential biomarker [69], while the application of isolated miR-18a inhibits CRC cell growth through the indirect regulation of the PI3K/AKT pathway [149] and has the potential to be sponged by the tumor-suppressing lncRNA CASC2 [150]. Similar to miR-21, miR-19a directly inhibits the tumor suppressor PTEN [151,152], as well as TIA1 [153], and can be predictive of the effectiveness of chemotherapeutic interventions on CRC [72,152], all while isolated miR-19a seemingly inhibits CRC angiogenesis via KRAS reduction [154]. Both miR-19a and miR-19b share identical seed sequences and miR-19b is also known to canonically inhibit PTEN expression [155,156], yet only miR-19b has been shown to inhibit tumor suppressor TP53 [157], further underscoring the effect of noncanonical miRNA influence. MiR-19b has already been indicated as a putative serum/plasma biomarker for other diseases, including lung cancer [158,159] and diabetic cardiomyopathy [160]. Similar to miR-21 and miR-19a, miR-92a activates the PI3K/AKT cell cycle pathway via PTEN inhibition, as well as activating Wnt/ß-catenin signaling, promoting carcinogenic development [59,161,162] and downregulating the tumor suppressors RECK and KLF4 [163,164]. Both miR-17 and miR-20a share an identical seed sequence and, therefore, share many similar confirmed targets in CRC (BCL2L11, CDKN1A, PTEN, TGFBR2, and VEGFA) [143,165], while also having individualized targets as seen with miR-17 and RBL2 and miR-20a with BID and SMAD4 [78,166], stressing a non-seed sequence-based influence. MiR-92a has already been put forward as a marker of CRC in both serum/plasma [63,80,81] and stool samples [54,167] and whose overexpression is a recognized marker of poor prognosis in CRC tissue [168]. Many studies point to dysregulated miRNAs, including the miR-17/92 cluster, as a predictor of chemotherapy efficacy in CRC patients [152,169,170]. Within the miR-17/92 cluster, there are contradictory results between studies that could be attributable to many issues, including, but not limited to, differences in the cellular context [165], the use of miRNAs in isolation [148,149,154] as opposed to the endogenous polycistronic expression of MIR17HG which increases CRC invasiveness [171], or structural components of pri-miR-19/72 itself that allow for the autoregulation of expression between each individual miRNA within the cluster [172,173,174]. As with miR-21, the miR-17/92 cluster is intimately tied to CRC at various levels of expression and continues to be an intense and promising area of study for biomarking CRC.

### 2.3. miR-143 and miR-145

Both miR-143 and miR-145 have different seed sequences and are found on chromosome 5 within the lncRNA CARMN locus [175], with miR-143 being slightly upstream of miR-145 and both under the control of the same SRF/MYOCD/NKX2.5 enhancer region [33]. The complete ablation of miR-143/145 results in the defective development of smooth muscle and aortic tissue [176]. While miR-21 and the miR-17/92 cluster are generally associated to be oncogenic in nature, both miR-143 and miR-145 are considered to be oncosuppressive, specifically within CRC [177,178,179], by mainly acting on the p53 and RAS/MAPK oncogenic pathways [180]. Both miR-143 and miR-145 are also known to directly downregulate CTNND1, which is a vital piece of Wnt/ß-catenin carcinogenesis [181,182]; thus, showing miR-143/145 are antioncogenic through multiple pathways. Originally thought to be highly expressed in intestinal epithelial cells, it was found that while miR-143/145 are vital to epithelial cell regeneration, their expression was overwhelmingly found in the mesenchyme [183,184]. The experimentally validated targets of miR-143 included reducing MAPK7 [185] and KRAS [177], both members of the RAS/MAPK signaling pathway. MiR-145 is known to canonically inhibit the well-recognized oncogene MYC [6], PI3K activator, IRS1 [186], and metastatic promoter FSCN1 [187]. A reduction in the antioncogenic effect of miR-143/145 can be further exacerbated by their sponging via ncRNAs [188,189,190,191]. The sequestration of miR-143/145 via ncRNA sponging adds to the notion that miRNA levels of expression are not absolute in their cellular or histological effects, as there is a need to consider the levels of other ncRNAs that may reduce or eliminate their efficacy. In isolation, reduced tissue levels of miR-143/145 seem to be a marker of CRC, and have been found to be a marker for large tumors [192], while not being diagnostic between clinical stages of malignancy [193]. A few studies, with lower-than-optimal patient numbers, have investigated fecal samples for levels of miR-143/145 and found a lowered expression in CRC patient samples [194,195], but further broader studies are required to confirm these results. In CRC patients, plasma/serum levels of miR-143/145 have been found to be upregulated [98,196], downregulated [74,85,86], or insignificant when compared to control samples [84]. These contradictory findings suggest that a reduced miR-143/145 is likely only a possible indicator of the presence of abnormal colorectal growth. However, the combined analysis of observed miR-143/145 levels, compared to known targets/sponges, as well as other known oncogenic ncRNAs may provide any avenue for more accurate conclusions of the relation between miR-143/145 and CRC.

### 2.4. miR-200 Family

The miR-200 family can be found across two chromosomes, with miR-200b, miR-200a, and miR-429 on chromosome 1 and miR-200c and miR-141 on chromosome 12 in the intergenic space between PTPN6 and PHB2. MiR-200a and miR-141 share identical seed sequences, while miR-200b, miR-200c, and miR-429 have the same seed sequence, with only a single letter difference between the two groups [197]. Early studies uncovered that canonical targets of the miR-200 family are ZEB1/ZEB2 transcripts [198,199], which are master regulators of the epithelial–mesenchymal transition (EMT) via the inhibition of E-cadherin expression [200,201], and ZEB proteins have a reciprocally negative feedback loop with the miR-200 family [202]. The inter-regulatory relationship of the miR-200 family downregulating ZEB1/ZEB2 that allows for the continued transcription of E-cadherin, which inhibits EMT, is generally accepted as the main carcinogenic pathway downregulated by the miR-200 family, specifically in CRC [203]. However, miR-200c specifically can increase the ability of metastasized CRC cells to proliferate in the liver [204], and the miR-200 family as a whole is oncogenic in some CRC cell lines [205], again emphasizing the temporal and site-specific nature of the consequences of miRNA activity. It is also known that the miR-200 family antagonizes tumor angiogenesis, as shown in several cancer models, through the targeting of the pro-inflammatory CXCL1 [206]. MiR-200 family expression can be seen as tumor-suppressive [198,207], while also promoting metastasis [99,208] and, thus, its relation to CRC is strong but ill-defined. Expanding on the tumor-suppressive nature of the miR-200 family, isolated miR-429 is known to suppress EMT through the downregulation of ONECUT2 [209]. MiR-141 on its own is also known to inhibit the translation of several tumor-suppressor genes [210,211,212]. As the miR-200 family is located at two distinct chromosomal locations, complete family in vivo knockout is difficult and has only very recently been accomplished in vitro with knockout cells showing increases in senescence and EMT signaling genes with similar expression patterns found in gastric cancer patient RNA-seq samples [213]. The abnormal regulation of the miR-200 family as a whole has been indicated as a marker for CRC in serum/plasma [208,214], and as individual miRNAs: miR-200a [215], miR-200b [89], miR-200c [86,97,98,99,100,101], miR-141 [89,98,99,101,216], and miR-429 [102]. Furthermore, elevated serum levels of miR-200c and miR-141 have been correlated with an increased CRC metastasis [97,99] and poor prognoses in CRC patients [97,99,100,101,216]. Lowered miR-200a and elevated miR-200b in serum show similar patterns of being correlated with poor outcomes in CRC patients [89,211]. Moreover, the effectiveness of the carcinogenic influence of each member of the miR-200 family can potentially be mitigated by other ncRNAs that competitively inhibit their actions through sponging [217,218,219,220]. As in the case of other miRNAs and CRC, the basic presence or absence of miR-200 family transcripts in the biofluids of CRC patients alone is not enough to be diagnostic and must be considered within a cellular and histological framework while also considering expression levels of other contextually relevant and impactful ncRNAs. 

### 2.5. miR-203

Within the genome, miR-203 is located intergenically between ASPG and KIF26A on chromosome 14. It first came to research prominence when it was found that its expression was high in colorectal adenocarcinomas compared to noncancerous tissue [221], yet other studies have found lowered miR-203 levels in similar CRC tissue [222,223]. Its expression is a key switch for the differentiation of basal skin epithelial cells by directly downregulating TP63 [224]. In this way, miR-203 acts as a tumor suppressor by directly inhibiting ΔNp63, which, when active, causes nuclear ß-catenin accumulation [225,226]. The whole-body knockout of miR-203 does not cause any overall developmental differences, but does invoke an expansion of proliferating keratinocytes, most notably during embryonic development [227]. In CRC cell lines, miR-203 reduction is required to maintain the stemness quality of cancerous lines [223,228,229], including through the reduction in the overexpressed and known CRC marker, NEDD9 [230,231]. Increases in miR-203 in CRC cells have been shown to canonically inhibit the oncogenic expression of AKT2, SIK2, CPEB4, EIF5A2, and TYMS [92,228,232,233,234,235]. Similar to other miRNAs, miR-203’s activity can be halted by means of sponging by ncRNAs, such as FBXL19-AS1 [236], BANCR [237], and LINC00657 [238]. Many studies into the serum/plasma levels of miR-203 in CRC patients have discovered that differential levels of miR-203 are most often an indicator of poor prognosis and increased metastasis usually marked by an increase in miR-203 [89,90,91,93,94,95,96], with some showing lowered levels [84,92]. A meta-analyses of 11 papers found a higher miR-203 expression in CRC tissue to be significant, and not in serum, but a combination of both serum and CRC tissue was predictive of a poor outcome [239]. Additionally, such as other miRNAs, miR-203 is known to increase chemosensitivity in CRC cells [232,235]. MiR-203 detection could present as a valuable piece of the puzzle in properly biomarking CRC. 

### 2.6. miR-135 Family

The miR-135 family is represented by miR-135a found just upstream of GLYCTK on chromosome 3, and miR-135b, contained within the genomic sequence of both the lncRNA BLACAT1 and the gene LEMD1 on chromosome 1. Both miR-135a and miR-135b share identical mature seed sequences and only one nucleotide difference in total. The miR-135 family was originally tagged as an oncogene by its targeting and downregulating of the tumor suppressor APC (with which miR-135b it has a reciprocally inhibitory relationship [240]) in CRC, which regulates the Wnt/ß-catenin pathway [241,242] as well as priming pancreatic cancer cells for pro-carcinogenic metabolic conditions through PFK1 downregulation [243]. In a CRC-specific context, miR-135a downregulates a metastasis driver MTSS1 [244] and miR-135b downregulates TGFBR2 [245], which is functionally mutated in up to 90% of CRC cases [246]. The effects of miR-135 can be antagonized through lncRNA sponging within a CRC context through circNOL10 [247]. Several studies have concentrated on modified levels of miR-135 family members in stool, tissue, and serum. As expected with the oncogenic nature of miR-135, stool samples show an increase in miR-135 expression [103,107] and even being predictive of later-stage (III-IV) cancer [104]. CRC tissue samples with an increased miR-135 family expression are indicative of poor prognosis and metastatic conditions [240,248,249,250]. Low serum levels of miR-135 have been correlated with CRC [106], while elevated levels have been able to elucidate differences between polyps and carcinomas [105], and, thus, the use of miR-135 as a serum biomarker has been controversial [111] and, thus, focusing on noninvasive fecal samples has been more productive and a better candidate for the detection of the miR-135 family.

### 2.7. miR-96 and miR-183

On chromosome 7, miR-96 is found in a cluster with miR-183 and upstream of NRF1. Research interest in miR-96/183-related pathologies and development began with discoveries that both miR-96 and miR-183 are vital to hair cell development and function in the inner ear [251,252], and increases in miR-96 and miR-183 are found within breast and lung cancer tissue while directly downregulating FOXO1 and FOXO3A [253,254,255,256], key tumor suppressors in the P13K/AKT pathway of carcinogenesis. The knockout of the miR-96/183 cluster causes increases in target SLC6A6, resulting in dysfunctional photoreception [257]. In vitro CRC experiments have shown that increases in miR-96 and miR-183 are associated with both an increased cell migration [258,259] and invasiveness [260,261]. Alongside this oncogenic character, a high miR-96 and miR-183 expression has also shown to increase resistance to chemotherapy treatments such as oxaliplatin [262], 5-fluoruracil [110], and radiation treatments [263], extending their oncogenic nature. Within studies of CRC tumor tissue samples, miR-96 and miR-183 were found to be elevated in all but one study [248,264,265,266,267], and that study found that lowered tumor tissue levels of miR-96 entailed poor patient prognoses [268]. A handful of studies have interrogated miR-96 levels in serum/plasma and have found that high levels [89,110], low levels [269], and no significant association with CRC [69] have been observed despite the fact that high miR-96 serum levels have been tied to both hepatocellular [269] and lung cancers [270]. Serum levels of miR-183 are consistently high in CRC patients [111,112]. The incongruous results from miR-96 could possibly be due to the neutralization of the oncogenic effects of miR-96 through sponging as has been seen in cervical cancer (STXBP5-AS1) [271] and pancreatic ductal adenocarcinoma (TP53TG1) [272]. In this vein, circ_0026344, has recently been shown to abrogate the oncogenic nature of miR-183 in CRC cells [273]. The consideration of miR-96 and miR-183 as biomarkers and targets for CRC diagnosis and treatment should, therefore, always be considered within the context of other mitigating ncRNAs.

### 2.8. miR-150

Just upstream of RPS11 on chromosome 19 is miR-150. Initial analyses of miR-150 found it to be vital to proper hematopoietic cell differentiation [274], specifically B cell maturation [275], through the downregulation of MYB, which was confirmed in CRC cells [276]. Unsurprisingly, the knockout of miR-150 results in B cell developmental difficulties coupled with obesity-related metabolic dysregulation [277]. In addition to downregulating MYB in CRC, miR-150 also directly lowers the expression of ß-catenin [278], VEGFA [279], and a known marker of poor prognoses in CRC patients, MUC4 [280]. As miR-150 canonically downregulates over several oncogenic pathways, it is generally seen as a tumor suppressor in CRC. In contrast, miR-150 is seen as oncogenic in gastric cancer tissues by downregulating the pro-apoptotic genes P2X_7_ [281] and EGR2 [282] with a single study showing increases in miR-150 in CRC tissue [283]. Resected tissue samples from CRC patients regularly show a significantly decreased expression of miR-150 when compared to both adenoma and healthy colonic tissue [278,279,284], and these reduced quantities can be indicative of poor prognoses [285]. However, when miR-150 expression was measured within serum samples of CRC patients, there have been observations of both increases [74,116] and decreases [64,113,114,115], yet again stressing the need for the analysis of miRNA levels within a context of known targets and other ncRNA inhibitors. In this regard, the growth-suppressing consequences of miR-150 expression in CRC can be arrested by means of sponging via the lncRNAs ZFAS1 [286] and PART1 [287]. The previous successes with detecting low levels of miR-150 in CRC patients’ serum suggests that the continued study of miR-150 and its relation to CRC, and known sequestering inhibitors, in serum/plasma will continue to be a fruitful area of investigation.

### 2.9. miR-195 

While a member of the miR-15 precursor family based on an identical seed sequence, miR-195 resides away from other family members and is found on chromosome 17 within the genomic location of lncRNA MIR497HG, just upstream of the 3′ UTR of C17ORF49. Modified levels of miR-195 first became an area of research in cardiac studies as increased levels of miR-195 were found to cause cardiac hypertrophy [288], as well as regulating cell cycle checkpoints in cardiomyocytes by targeting CHEK1 [289]. In terms of cancer research, early studies found that miR-195 was downregulated in hepatocellular cancer tissue via microarray [290], and that miR-195 inhibited CCND1 translation in both hepatocellular and colorectal cell lines [291], halting the cell cycling necessary for carcinogenic progression. Recently, miR-195 has also been found to be important in the cellular maintenance of the blood–brain barrier [292]. In addition to inhibiting CHEK1 in CRC cells, miR-195 prevents WNT3A translation [293], which is known to activate the Wnt/ß-catenin oncogenic signaling pathway in CRC cells [294]. MiR-195 also canonically inhibits NOTCH2 [295] and BCL2L11 [296] in CRC cells, providing more pathways of tumor suppression through preventing EMT and promoting apoptosis, respectively. Similar to in vitro studies, miR-195 is consistently downregulated miRNA in CRC tissue samples [117,295,296,297,298,299,300,301], and the addition of miR-195 makes CRC cells sensitized to currently used chemotherapy interventions such as 5-fluorouracil [298,302], doxorubicin [303], and radiation therapy [304]. Surprisingly few studies have focused on serum/plasma levels of miR-195 in CRC patients, but results have shown the expected lowered levels of miR-195 [117]. Just as observed with other CRC-associated miRNAs, the tumor-suppressing actions of miR-195 can be annulled when sequestered through sponging by highly expressed lncRNA/circRNAs, with all of them promoting CRC carcinogenesis and progression [305,306,307,308]. Being able to couple known miR-195 targets and sponges, with stably reduced miR-195 in CRC tissues insinuates that miR-195 is a ripe area of further research in CRC.

## 3. Discussion

### miRNAs, ceRNAs, and Chemosensitivity in CRC: A War of Attrition

Once the ability for miRNAs to be sponged out of efficacy by other ceRNAs was found to be robust, this provided the incentive to construct synthetic ceRNAs as sponges for oncogenic miRNAs. Methodologies for the building of unique miRNA sponges with multiple seed target sites were quickly introduced [309]. In practice, these sponges have indeed been able to isolate oncogenic miRNAs such as that observed with miR-21 sponging in renal cancer [310], gastric cancer [311], glioblastomas [312], and esophageal carcinomas [313]. Sponging has also been shown to be a direct driver of metastasis in CRC by circHIPK3 isolating miR-7 [314]. Thus, sponging miRNAs, such as the presence of miRNAs themselves, can be both oncogenic and suppressive in CRC. One very interesting operation of miRNAs in CRC is the ability to either sensitize or desensitize tumor cells to established chemotherapies and, therefore, be predictive of patient outcomes [170]. Both increased miR-21 and miR-17/92 expressions are indicative of chemoresistant tumors [110,169], and, thus, utilizing either increased ceRNA or synthetic ncRNA sponges would be ideal in helping sensitize patients to previously resistant chemotherapy or radiation. In contrast, miRNAs such as miR-203, whose increased expression is correlated with chemotolerance in CRC [232,235], represent a ripe opportunity to be used as an adjuvant to make resistant CRC cells susceptible to previously ineffective treatments. These examples epitomize the possible capabilities of targeting miRNA and miRNA-affected pathways in CRC treatments.

Despite observed patterns, one of the most consistent features of biofluid-borne ncRNAs in CRC patients is that their levels are almost never entirely correlative with disease presence, stage of development, metastasis extent, or susceptibility to known treatments. The colon and rectum are incredibly dynamic environments that constantly respond to various environmental cues, nutritional input, hypoxic conditions, and microbial influence, both commensal and pathogenic. This regularly reactive state causes the rapidly modulable nature of RNA to be an ideal molecule for responding to these shifting settings. This can manifest in the inducible power of miRNAs to alter mRNA translation, as well as ceRNAs’ ability to negate those same miRNAs’ effect on mRNA through competitive sequestration. Early studies on miRNAs and CRC quickly focused on elucidating their relationship to effectors of known carcinogenic pathways, such as mRNAs involved in the Wnt/ß-catenin, PI3K/AKT, p53/c-MYC, and MAPK/ERK pathways. More recent analyses have unveiled the capacity for ceRNAs to mitigate the influence of miRNAs in CRC (Table 2). Neither the isolated investigation of miRNA on mRNA, or ceRNA on miRNA influence would complete the carcinogenic picture due to their attritive relationships. 

The molecular CRC environment becomes further complicated as modifications to miRNA and ceRNA are known to alter their effect such as A-to-I editing [355] or m6A methylation [356]. This struggle between mRNA, miRNA, and ceRNAs over the carcinogenic capacity within CRC tumors demands the need for a more comprehensive review of the overall RNA influence in personalized CRC diagnosis and treatment. The main cytological affectations of miRNAs cannot be fully explained except in light of taking downstream targets into account. For example, circACAP2 is capable of sponging both oncogenic miR-21 [317] and suppressive miR-143 [330] and, therefore, circACAP2 in isolation could be considered carcinogenic or suppressive in relation to the sequestration of miR-21 or miR-143 in CRC. Thus, the vital examinations of interactive miRNA/ceRNA networks within a CRC context are becoming more necessary and prevalent [357,358,359], including studying the epigenetic regulation of miRNA/ceRNA expression [360], instead of focusing on individual cases of ceRNA/miRNA/mRNA interaction in order to provide a more thorough exploration of their collective effect on CRC. These examinations take a wider view of the interactions of multiple RNA species as they operate as a collective axis instead of concentrating on individual RNA species and allow for more personalized medical strategies [361]. Preliminary patterns of ceRNA-miRNA-mRNA networks in CRC samples are beginning to be discovered and provide a better framework for understanding the heterogenous CRC environment [357,362,363], and should be used as templates for the integrative ncRNA analysis in CRC patients. Figure 1 summarizes ceRNA-miRNA-mRNA networks in CRC, in which oncogenic or tumor suppressive miRNAs are dysregulated. The dysregulated miRNAs and ceRNAs can be potentially developed as biomarkers in the diagnosis of CRC as well as therapeutic targets to treat the disease. Continued work into personalized ncRNA-based diagnoses/treatments for the multifaceted and insidious nature of CRC could provide promising new avenues of prevention, diagnosis, and treatment.

## Figures and Tables

**Figure 1 ijms-23-02166-f001:**
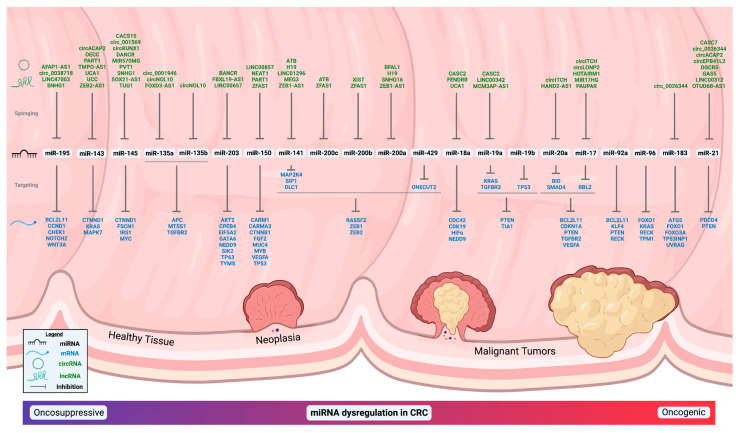
Dynamic relationships between miRNAs and ceRNAs and their connection to CRC. The miRNAs are listed in general/relative order of oncosuppressive to oncogenic in CRC from left to right with miRNA with dual natures being more central. All miRNAs, “sponging” ceRNAs (circRNA and lncRNA), and mRNAs listed have references cited within the main text and/or tables.

**Table 1 ijms-23-02166-t001:** Dysregulated miRNAs detected in noninvasive biofluids in CRC patient samples.

miRNA	Biofluid Source	Levels in CRC	Reference
miR-21	Stool	Upregulated	[54]
	Stool	Upregulated	[55]
	Stool	Upregulated	[56]
	Serum	Upregulated	[57]
	Serum	Upregulated	[58]
	Serum	Upregulated	[59]
	Serum	Upregulated	[60]
	Serum	Upregulated	[61]
	Serum	Upregulated	[62]
	Serum	Upregulated	[63]
	Serum	Upregulated	[64]
miR-17	Serum	Upregulated	[65]
	Serum	Upregulated	[66]
	Serum	Upregulated	[67]
	Serum	Upregulated	[62]
miR-18a	Plasma	Downregulated	[68]
	Serum	Upregulated	[69]
	Serum	Upregulated	[70]
miR-19a	Serum	Upregulated	[65]
	Serum	Upregulated	[71]
	Serum	Upregulated	[43]
	Serum	Upregulated	[72]
	Serum	Upregulated	[73]
	Serum	Upregulated	[74]
	Serum	Upregulated	[61]
	Serum	Upregulated	[75]
	Serum	Upregulated	[76]
miR-20a	Serum	Downregulated	[77]
	Serum	Upregulated	[78]
	Serum	Upregulated	[62]
	Serum	Upregulated	[65]
	Serum	Upregulated	[74]
miR-19b	Serum	Upregulated	[43]
miR-92a	Plasma	Upregulated	[79]
	Stool	Upregulated	[54]
	Serum	Upregulated	[75]
	Serum	Upregulated	[80]
	Serum	Upregulated	[63]
	Serum	Upregulated	[81]
	Serum	Upregulated	[82]
	Serum	Upregulated	[83]
	Serum	Upregulated	[84]
miR-143	Plasma	Downregulated	[85]
	Serum	Downregulated	[74]
	Serum	Downregulated	[86]
	Serum	Upregulated	[87]
	Serum	No Significant Change	[84]
miR-145	Plasma	Downregulated	[85]
	Serum	Downregulated	[88]
	Serum	Downregulated	[74]
	Serum	Downregulated	[86]
	Serum	No Significant Change	[84]
	Serum	Downregulated	[66]
miR-203	Plasma	Upregulated	[89]
	Serum	Downregulated	[84]
	Serum	Upregulated	[90]
	Serum	Upregulated	[91]
	Serum	Downregulated	[92]
	Serum	Upregulated	[93]
	Serum	Upregulated	[94]
	Serum	Upregulated	[95]
	Serum	Upregulated	[96]
miR-200a	None Reported		
miR-200b	Plasma	Upregulated	[89]
miR-200c	Serum	Upregulated	[97]
	Serum	Downregulated	[86]
	Serum	Upregulated	[98]
	Serum	Upregulated	[99]
	Serum	Upregulated	[100]
	Serum	Upregulated	[101]
miR-141	Plasma	Upregulated	[89]
	Serum	Upregulated	[98]
	Serum	Upregulated	[99]
	Serum	Upregulated	[101]
miR-429	Serum	Upregulated	[102]
miR-135a	Stool	Upregulated	[103]
	Stool	Upregulated	[104]
	Serum	Upregulated	[105]
	Serum	Downregulated	[106]
miR-135b	Stool	Upregulated	[107]
	Stool	Upregulated	[108]
	Serum	Upregulated	[109]
miR-96	Plasma	Upregulated	[89]
	Serum	Upregulated	[110]
	Serum	No Significant Change	[69]
miR-183	Serum	Upregulated	[111]
	Serum	Upregulated	[112]
miR-150	Serum	Downregulated	[113]
	Serum	Upregulated	[74]
	Serum	Downregulated	[114]
	Serum	Downregulated	[115]
	Serum	Upregulated	[116]
	Serum	Downregulated	[64]
miR-195	Plasma	Downregulated	[117]
	Serum	Downregulated	[118]

**Table 2 ijms-23-02166-t002:** Reported ceRNAs and targeted miRNAs in CRC.

miRNA	ceRNA in CRC	Reference
miR-21	CASC7	[315]
	circ_0026344	[316]
	circACAP2	[317]
	circEPB41L2	[318]
	DGCR5	[319]
	GAS5	[36]
	LINC00312	[320]
	OTUD6B-AS1	[321]
miR-17	circITCH	[12]
	circLONP2	[322]
	HOTAIRM1	[11]
	MIR17HG	[171]
	PAUPAR	[323]
miR-18a	CASC2	[150]
	FENDRR	[324]
	UCA1	[325]
miR-19a	CASC2	[326]
	LINC00342	[327]
	MCM3AP-AS1	[328]
miR-20a	circITCH	[12]
	HAND2-AS1	[329]
miR-19b	None Reported	
miR-92a	None Reported	
miR-143	circACAP2	[330]
	OECC	[331]
	PART1	[190]
	TMPO-AS1	[332]
	UCA1	[189]
	UCC	[188]
	ZEB2-AS1	[333]
miR-145	CACS15	[334]
	circ_001569	[191]
	circRUNX1	[335]
	DANCR	[336]
	MIR570MG	[337]
	PVT1	[338]
	SNHG1	[339]
	SOX2-AS1	[340]
	TUG1	[341]
miR-203	BANCR	[237]
	FBXL19-AS1	[236]
	LINC00657	[238]
miR-200a	BFAL1	[342]
	H19	[217]
	SNHG16	[343]
	ZEB1-AS1	[344]
miR-200b	XIST	[345]
	ZFAS1	[218]
miR-200c	ATB	[346]
	ZFAS1	[218]
miR-141	ATB	[347]
	H19	[348]
	LINC01296	[349]
	MEG3	[350]
	ZEB1-AS1	[218]
miR-429	None Reported	
miR-135a	circ_0001946	[351]
	circNOL10	[247]
	FOXD3-AS1	[352]
miR-135b	circNOL10	[247]
miR-96	None Reported	
miR-183	circ_0026344	[273]
miR-150	LINC00857	[353]
	NEAT1	[354]
	PART1	[287]
	ZFAS1	[286]
miR-195	AFAP-AS1	[308]
	circ_0038718	[307]
	LINC00473	[306]
	SNHG1	[305]

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
