# Peer review of "MicroRNAs and ‘Sponging’ Competitive Endogenous RNAs Dysregulated in Colorectal Cancer: Potential as Noninvasive Biomarkers and Therapeutic Targets"

_ijms, 2022, doi:10.3390/ijms23042166_

Round 1
Reviewer 1 Report
The manuscript entitled "microRNAs and ‘sponging’ competitive endogenous RNAs dysregulated in colorectal cancer: potential as non-invasive biomarkers and therapeutic targets". The author described the miRNA biomarkers signatures found within the biofluids of the colorectal cancer subjects. The article would be a great deal of interest for researchers who are interested in designing studies related to colorectal cancer. It is suggested to the authors to make changes to Figure 1 (page 5) kindly add a picture with good clarity. Best
Author Response
We sincerely appreciate the time and supportive words from the reviewer. We have attached a high quality PDF file (300 dpi) of our figure to this review response. Additionally, we will be attaching a JPEG and PNG file of the same figure at high resolution (300 dpi) that cannot be attached in this review response. We anticipate that the use of any of the attached files will be of appropriate quality for publication.

Reviewer 2 Report
Accept after minor revision.
Author Response
We'd like to thank the reviewer for taking the time to assess the manuscript. Spelling check and overall English editing have been completed. We have attached a PDF file of our Figure 1 at high resolution (300 dpi). Additionally, we have attached to the main manuscript both a JPEG and PNG (300 dpi) to ensure that Figure 1 is in any necessary formatting.

Reviewer 3 Report
This review article seems me of interest but the presentation is not up to the mark. The main drawback with this article is lack of critical evaluation and presetting the authors views. The quality of Figures are not good, for example I am not able to read some text in Figure 1. Other important suggestions are –
- Improve English throughout the manuscript.
- potential should be Potential in the tittle.
- Provide quantitative information in the abstract.
- The concussion should be concise and to the points indicating the application of the work.
- Write one paragraph in the introduction on cancer in general with citations of the following refs.
Med. Chem. Res., 22 (3), 1386-1398 (2013).; Future Med. Chem., 5 (2), 135-146 (2013); Curr. Pharmaceut. Anal., 1 (1), 109-125(2005).
Author Response
1. Improve English throughout the manuscript
With respect to the English employed throughout the manuscript, we do not deem the vocabulary, syntax, spelling, organization or nomenclature to be lacking or in need of drastic change. The manuscript has been evaluated for proper English by both human and word processor editors. If the reviewer has specific/individual instances that are seen to be needing improvement, we would be highly receptive to these comments as this advice is ambiguous. If the editors of IJMS decide that an English editing is necessary, we would be happy to have that accomplished.
2. potential should Potential in the title
A very perceptive observation. The prescribed edit has been adopted into the manuscript title.
3. Provide quantitative information in the abstract
As fellow professional scientists, we agree than quantitative data is paramount and we will happily add it, as needed. However, we are unsure what quantitative data the reviewer is suggesting should be added to the abstract of this review. Amongst a few possibilities could be:
- Percentages of colonoscopies that successfully diagnose colorectal cancer
- Ratio of colorectal cancers than can or cannot be properly diagnosed through colonoscopies
- Success rate of chemotherapy/radiation on colorectal cancer so as to present the necessity for novel miRNA adjuvants and/or therapeutics
- Number of miRNA, ceRNA drugs currently FDA approved and in use
- Numbers of miRNA currently in clinical trials for colorectal cancer
Any suggestions or clarity on other quantitative reference points would be greatly appreciated.
4. The concussion should be concise and to the points indicating the application of the work
We believe that the reviewer is intending to draw our attention to our discussion section and their use of the word concussion is in error. Within our discussion section, we followed the IJMS guidelines for discussion sections for Research Manuscript Sections under Manuscript Preparation (According to IJMS guidelines for review papers, discussion sections are not required):
“Discussion: Authors should discuss the results and how they can be interpreted in perspective of previous studies and of the working hypotheses. The findings and their implications should be discussed in the broadest context possible and limitations of the work highlighted. Future research directions may also be mentioned. This section may be combined with Results.”
It is our contention that our discussion section fulfills the direction of these guidelines by synthesizing our contention that the integration of miRNA-ceRNA-mRNA networks would likely be more accurate, precise, and personal to the individual colorectal cancer patient as compared to measuring these RNA species in isolation which has previously been accomplished. We suggest that these networks should be developed into personalized RNA adjuvant, diagnostic, or therapeutic interventions. For the sake of succinctness, our discussion section could have potentially extraneous language, used to contextualize our suggestions, removed. Another edit could be the simple removal of the title “Discussion” while keeping the subheadings.
5. Write one paragraph in the introduction on cancer in general with citations of the following refs.
Med. Chem. Res., 22 (3), 1386-1398 (2013).; Future Med. Chem., 5 (2), 135-146 (2013); Curr. Pharmaceut. Anal., 1 (1), 109-125(2005).
Thank you for the astute comment on adding more information on cancer, in a general sense. We agree that a brief addition of information regarding the cellular origin and nature of cancer as a disease would be beneficial to the manuscript. The following additional text and references have been added:
“Colorectal cancer (CRC) most commonly manifests as neoplasias consisting mainly of mutated intestinal epithelial cells that are unable to maintain proper differentiation status and/or connections to neighboring cells, leading to unchecked cellular division and dysregulation of oncogenic and/or tumor suppressive genes [13,14]. Most CRC cases do not yet have a prescribed causative provenance and are thus considered sporadic (70%) [15].”
[13] Huels, D.J.; Sansom, O.J. Stem vs non-stem cell origin of colorectal cancer. Br. J. Cancer 2015, 113, 1–5, doi:10.1038/bjc.2015.214.
[14] Blanpain, C. Tracing the cellular origin of cancer. Nat. Cell Biol. 2013, 15, 126–134, doi:10.1038/ncb2657.
[15] Mármol, I.; Sánchez-de-Diego, C.; Dieste, A.P.; Cerrada, E.; Yoldi, M.J.R. Colorectal carcinoma: A general overview and future perspectives in colorectal cancer. Int. J. Mol. Sci. 2017, 18, doi:10.3390/ijms18010197.
Unfortunately, we do not believe that the publications that the reviewer suggested are of useful relevance to our manuscript as they are not focused on colorectal cancer or any ncRNA species and therefore have not been included. These publications could have been suggested by the reviewer in order to increase their citation number rather than as relevant references to the subject itself.
Finally, we have attached a PDF file of our Figure 1 at high resolution (300 dpi). Additionally, we have attached to the main manuscript both a JPEG and PNG (300 dpi) to ensure that Figure 1 is in any necessary formatting.

Round 2
Reviewer 3 Report
Accept